# EXTENDING THE FRAMEWORK OF EQUILIBRIUM PROPAGATION TO GENERAL DYNAMICS

## ABSTRACT

The biological plausibility of the backpropagation algorithm has long been doubted by neuroscientists. Two major reasons are that neurons would need to send two different types of signal in the forward and backward phases, and that pairs of neurons would need to communicate through symmetric bidirectional connections. We present a simple two-phase learning procedure for fixed point recurrent networks that addresses both these issues. In our model, neurons perform leaky integration and synaptic weights are updated through a local mechanism. Our learning method extends the framework of Equilibrium Propagation to general dynamics, relaxing the requirement of an energy function. As a consequence of this generalization, the algorithm does not compute the true gradient of the objective function, but rather approximates it at a precision which is proven to be directly related to the degree of symmetry of the feedforward and feedback weights. We show experimentally that the intrinsic properties of the system lead to alignment of the feedforward and feedback weights, and that our algorithm optimizes the objective function.

## 1 INTRODUCTION

Deep learning (LeCun et al., 2015) is the de-facto standard in areas such as computer vision (Krizhevsky et al., 2012), speech recognition (Hinton et al., 2012) and machine translation (Bahdanau et al., 2015). These applications deal with different types of data and share little in common at first glance. Remarkably, all these models typically rely on the same basic principle: optimization of objective functions using the *backpropagation* algorithm. Hence the question: does the cortex in the brain implement a mechanism similar to backpropagation, which optimizes objective functions?

The backpropagation algorithm used to train neural networks requires a side network for the propagation of error derivatives, which is vastly seen as biologically implausible (Crick, 1989). One hypothesis, first formulated by Hinton & McClelland (1988), is that error signals in biological networks could be encoded in the temporal derivatives of the neural activity and propagated through the network via the neuronal dynamics itself, without the need for a side network. Neural computation would correspond to both inference and error back-propagation. This work also explores this idea.

The framework of Equilibrium Propagation (Scellier & Bengio, 2017) requires the network dynamics to be derived from an energy function, enabling computation of an exact gradient of an objective function. However, in terms of biological realism, the requirement of symmetric weights between neurons arising from the energy function is not desirable. The work presented here extends this framework to general dynamics, without the need for energy functions, gradient dynamics, or symmetric connections.

Our approach is the following. We start from classical models in neuroscience for the dynamics of the neuron's membrane voltage and for the synaptic plasticity (section 3). Unlike in the Hopfield model (Hopfield, 1984), we do not assume pairs of neurons to have symmetric connections. We then describe an algorithm for supervised learning based on these models (section 4) with minimal extra assumptions. Our model is based on two phases: at prediction time, no synaptic changes occur, whereas a local update rule becomes effective when the targets are observed. The proposed update mechanism is compatible with spike-timing-dependent plasticity (Bengio et al., 2017), which supposedly governs synaptic changes in biological neural systems. Finally, we show that the proposed algorithm has the desirable machine learning property of optimizing an objective function (section 5). We show this experimentally (Figure 3) and we provide the beginning for a theoretical explanation.

## 2 MOVING BEYOND ENERGY-BASED MODELS AND GRADIENT DYNAMICS

Historically, models based on energy functions and/or gradient dynamics have represented a key subject of neural network research. Their mathematical properties often allow for a simplified analysis, in the sense that there often exists an elegant formula or algorithm for computing the gradient of the objective function (Ackley et al., 1985; Movellan, 1990; Scellier & Bengio, 2017). However, we argue in this section that

1. due to the energy function, such models are very restrictive in terms of dynamics they can model - for instance the Hopfield model requires symmetric weights,

2. machine learning algorithms do not require computation of the gradient of the objective function, as shown in this work and the work of Lillicrap et al. (2016).

In this work, we propose a simple learning algorithm based on few assumptions. To this end, we relax the requirement of the energy function and, at the same time, we give up on computing the gradient of the objective function.

We believe that, in order to make progress in biologically plausible machine learning, dynamics more general than gradient dynamics should be studied.

As discussed in section 6, another motivation for studying more general dynamics is the possible implementation of machine learning algorithms, such as our model, on analog hardware: analog circuits implement differential equations, which do not generally correspond to gradient dynamics.

### 2.1 GRADIENT DYNAMICS ARE NOT GENERIC DYNAMICS

Most dynamical systems observed in nature cannot be described by gradient dynamics. A gradient field is a very special kind of vector field, precisely because it derives from a primitive scalar function. The existence of a primitive function considerably limits the "number of degrees of freedom" of the vector field and implies important restrictions on the dynamics.

In general, a vector field does not derive from a primitive function. In particular, the dynamics of the leaky integrator neuron model studied in this work (Eq. 1) is not a gradient dynamics, unless extra (biologically implausible) assumptions are made, such as exact symmetry of synaptic weights ($W_{ij} = W_{ji}$) in the case of the Hopfield model.

### 2.2 MACHINE LEARNING DOES NOT REQUIRE GRADIENT COMPUTATION

Machine learning relies on the basic principle of optimizing objective functions. Most of the work done in deep learning has focused on optimizing objective functions by gradient descent in the weight space (thanks to backpropagation). Although it is very well known that following the gradient is not necessarily the best option – many optimization methods based on adaptive learning rates for individual parameters have been proposed such as RMSprop Tieleman & Hinton (2012) and Adagrad Duchi et al. (2011) – almost all proposed optimization methods rely on *computing* the gradient, even if they do not *follow* the gradient. In the field of deep learning, "computing the gradient" has almost become synonymous with "optimizing".

In fact, in order to optimize a given objective function, not only following the gradient unnecessary, but one does not even need to *compute* the gradient of that objective function. A weaker sufficient condition is to compute a direction in the parameter space whose scalar product with the gradient is negative, without computing the gradient itself.

A major step forward was achieved by Lillicrap et al. (2016). One of the contributions of their work was to dispel the long-held assumption that a learning algorithm should compute the gradient of an objective function in order to be sound. Their algorithm computes a direction in the parameter space that has at first sight little to do with the gradient of the objective function. Yet, their algorithm "learns" in the sense that it optimizes the objective function. By giving up on the idea of computing the gradient of the objective function, a key aspect rendering backpropagation biologically implausible could be fixed, namely the weight transport problem.

The work presented here is along the same lines. We give up on the idea of computing the gradient of the objective function, and by doing so, we get rid of the biologically implausible symmetric connections required in the Hopfield model. In this sense, the "weight transport" problem in the

backpropagation algorithm appears to be similar, at a high level, to the requirement of symmetric connections in the Hopfield model.

We suggest that in order to make progress in biologically plausible machine learning, it might be necessary to move away from computing the true gradients in the weight space. An important theoretical effort to be made is to understand and characterize the dynamics in the weight space that optimize objective functions. The set of such dynamics is of course much larger than the tiny subset of gradient dynamics.

## 3 CLASSICAL DYNAMICS IN NEUROSCIENCE

We denote by $s_i$ the averaged membrane voltage of neuron $i$ across time, which is continuous-valued and plays the role of a state variable for neuron $i$. We also denote by $\rho(s_i)$ the firing rate of neuron $i$. We suppose that $\rho$ is a deterministic function (nonlinear activation) that maps the averaged voltage $s_i$ to the firing rate $\rho(s_i)$. The synaptic strength from neuron $j$ to neuron $i$ is denoted by $W_{ij}$.

### 3.1 LEAKY INTEGRATOR NEURON MODEL

In biological neurons a classical model for the time evolution of the membrane voltage $s_i$ is the rate-based leaky integrator neuron model, in which neurons are seen as performing leaky temporal integration of their past inputs Dayan & Abbott (2001):

$$\frac{ds_i}{dt} = \sum_j W_{ij}\rho(s_j) - s_i. \tag{1}$$

Unlike energy-based models such as the Hopfield model (Hopfield, 1984) that assume symmetric connections between neurons, in the model studied here the connections between neurons are not tied. Thus, our model is described by a directed graph, whereas the Hopfield model is best regarded as an undirected graph (Figure 1).

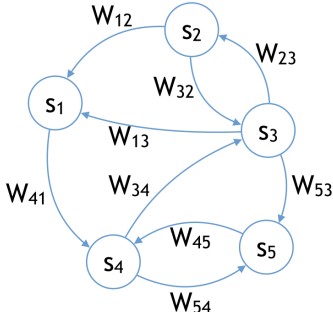
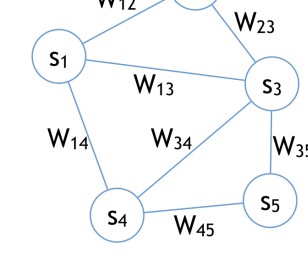

(a) The network model studied here is best represented by a directed graph.

(b) The Hopfield model is best represented by an undirected graph.

Figure 1: From the point of view of biological plausibility, the symmetry of connections in the Hopfield model is a major drawback (1b). The model that we study here is, like a biological neural network, a directed graph (1a).

### 3.2 SPIKE-TIMING DEPENDENT PLASTICITY

Spike-Timing Dependent Plasticity (STDP) is considered a key mechanism of synaptic change in biological neurons (Markram & Sakmann, 1995; Gerstner et al., 1996; Markram et al., 2012). STDP is often conceived of as a spike-based process which relates the change in the synaptic weight $W_{ij}$ to the timing difference between postsynaptic spikes (in neuron $i$) and presynaptic spikes (in neuron $j$) (Bi & Poo, 2001). In fact, both experimental and computational work suggest that postsynaptic voltage, not postsynaptic spiking, is more important for driving LTP (Long Term Potentiation) and LTD (Long Term Depression) (Clopath & Gerstner, 2010; Lisman & Spruston, 2010).

Similarly, Bengio et al. (2017) have shown in simulations that a simplified Hebbian update rule based on pre- and post-synaptic activity can functionally reproduce STDP:

$$dW_{ij} \propto \rho(s_j)ds_i. \tag{2}$$

Throughout this paper we will refer to this update rule (Eq. 2) as "STDP-compatible weight change" and propose a machine learning justification for such an update rule.

### 3.3 VECTOR FIELD $\mu$ IN THE STATE SPACE

Let $s = (s_1, s_2, \dots)$ be the global state variable and parameter $W$ the matrix of connection weights $W_{ij}$. We write $\mu(W, s)$ the vector whose components are defined as

$$\mu_i(W, s) := \sum_j W_{ij} \rho(s_j) - s_i \tag{3}$$

defining a vector field over the neurons state space, indicating in which direction each neuron's activity changes:

$$\frac{ds}{dt} = \mu(W, s). \tag{4}$$

Since $\rho(s_j) = \frac{\partial \mu_i}{\partial W_{ij}}(W, s)$, the weight change Eq. 2 can also be expressed in terms of $\mu$ in the form $dW_{ij} \propto \frac{\partial \mu_i}{\partial W_{ij}}(W, s) ds_i$. Note that for all $i' \neq i$ we have $\frac{\partial \mu_{i'}}{\partial W_{ij}} = 0$ since to each synapse $W_{ij}$ corresponds a unique post-synaptic neuron $s_i$. Hence $dW_{ij} \propto \frac{\partial \mu}{\partial W_{ij}}(W, s) \cdot ds$. We rewrite the STDP-compatible weight change in the more concise form

$$dW \propto \frac{\partial \mu}{\partial W}(W, s) \cdot ds. \tag{5}$$

## 4 A BIOLOGICALLY PLAUSIBLE LEARNING ALGORITHM FOR FIXED POINT RECURRENT NETWORKS WITHOUT TIED WEIGHTS

The framework and the algorithm in their general forms are described in Appendix A.

To illustrate our algorithm, we consider here the supervised setting in which we want to predict an output y given an input x. We describe a simple two-phase learning procedure based on the dynamics Eq. 4 and Eq. 5 for the state and the parameter variables. This algorithm is similar to the one proposed by Scellier & Bengio (2017), but here we do not assume symmetric weights between neurons. Note that similar algorithms have also been proposed by O'Reilly (1996); Hertz et al. (1997) or more recently by Mesnard et al. (2016). Our contribution in this work are theoretical insights into why the proposed algorithm works.

### 4.1 TRAINING OBJECTIVE

In the supervised setting studied here, the units of the network are split in two sets: the inputs x whose values are always clamped, and the dynamically evolving units $h$ (the neurons activity, indicating the state of the network), which themselves include the hidden layers ($h_1$ and $h_2$ here) and an output layer ($h_0$ here), as in Figure 2. In this context the vector field $\mu$ is defined by its components $\mu_0$, $\mu_1$ and $\mu_2$ on $h_0$, $h_1$ and $h_2$ respectively, as follows:

$$\mu_0(W, \mathrm{x}, h) = W_{01} \cdot \rho(h_1) - h_0, \tag{6}$$
$$\mu_1(W, \mathrm{x}, h) = W_{12} \cdot \rho(h_2) + W_{10} \cdot \rho(h_0) - h_1, \tag{7}$$
$$\mu_2(W, \mathrm{x}, h) = W_{23} \cdot \rho(\mathrm{x}) + W_{21} \cdot \rho(h_1) - h_2. \tag{8}$$

Here the scalar function $\rho$ is applied elementwise to the components of the vectors. The neurons $h$ follow the dynamics

$$\frac{dh}{dt} = \mu(W, \mathrm{x}, h). \tag{9}$$

In this section and the next we use the notation $h$ rather than $s$ for the state variable.

The layer $h_0$ plays the role of the output layer where the prediction is read. The target outputs, denoted by y, have the same dimension as the output layer $h_0$. The discrepancy between the output units $h_0$ and the targets y is measured by the quadratic cost function

$$C(h, \mathrm{y}) := \frac{1}{2} \|\mathrm{y} - h_0\|^2. \tag{10}$$

Unlike in the continuous Hopfield model, here the feed-forward and feedback weights are not tied, and in general the state dynamics Eq. 9 is not guaranteed to converge to a fixed point. However we observe experimentally that the dynamics almost always converges. We will see in section 5 that, for a whole set of values of the weight matrix $W$. the dynamics of the neurons $h$ converges. Assuming this condition to hold, the dynamics of the neurons converge to a fixed point which we denote by $h^0$ (beware not to confuse with the notation for the output units $h_0$). The prediction $h_0^0$ is then read out on the output layer and compared to the actual target y. The objective function (for a single training case $(\mathrm{x}, \mathrm{y})$) that we aim to minimize is the cost at the fixed point $h^0$, which we write

$$J := C\left(h^0, \mathrm{y}\right). \tag{11}$$

Note that this objective function is the same as the one proposed by Almeida (1987); Pineda (1987). Their method to optimize $J$ is to compute the gradient of $J$ thanks to an algorithm which they call "Recurrent Backpropagation". Other methods related to Recurrent Backpropagation exist to compute the gradient of $J$ - in particular the "adjoint method", "implicit differentiation" and "Backprop Through Time". These methods are biologically implausible, as argued in Appendix B.

Here our approach to optimize $J$ is to give up on computing the true gradient of $J$ and, instead, we propose a simple algorithm based only on the leaky integrator dynamics (Eq. 4) and the STDP-compatible weight change (Eq. 5). We will show in section 5 that our algorithm computes a proxy for the gradient of $J$. Also, note that in its general formulation, our algorithm applies to any vector field $\mu$ and cost function $C$ (Appendix A)

## 4.2 EXTENDED DYNAMICS

The idea of Equilibrium Propagation (Scellier & Bengio, 2017) is to see the cost function $C$ (Eq. 10) as an external potential energy for the output units $h_0$, which can drive them towards their target y. Following the same idea we define the "extended vector field" $\mu^\beta$ as

$$\mu^\beta := \mu - \beta\frac{\partial C}{\partial h}, \tag{12}$$

and we redefine the dynamics of the state variable $h$ as

$$\frac{dh}{dt} = \mu^\beta(W, \mathrm{x}, h, \mathrm{y}). \tag{13}$$

The real-valued scalar $\beta \geq 0$ controls whether the output $h_0$ is pushed towards the target y or not, and by how much. We call $\beta$ the "influence parameter" or "clamping factor".

The differential equation of motion Eq. 13 can be seen as a sum of two "forces" that act on the temporal derivative of the state variable $h$. Apart from the vector field $\mu$ that models the interactions between neurons within the network, an "external force" $-\beta\frac{\partial C}{\partial h}$ is induced by the external potential $\beta C$ and acts on the output neurons:

$$-\beta\frac{\partial C}{\partial h_0} = \beta(\mathrm{y} - h_0), \tag{14}$$

$$-\beta\frac{\partial C}{\partial h_i} = 0, \qquad \forall i \geq 1. \tag{15}$$

The form of Eq. 14 suggests that when $\beta = 0$, the output units $h_0$ are not sensitive to the targets y from the outside world. In this case we say that the network is in the *free phase* (or first phase). When $\beta > 0$, the "external force" drives the output units $h_0$ towards the target y. When $\beta \gtrsim 0$ (small positive value), we say that the network is in the *weakly clamped phase* (or second phase). Also, note that the case $\beta \to \infty$, not studied here, would correspond to fully clamped outputs.

## 4.3 TWO-PHASE ALGORITHM AND BACKPROPAGATION OF ERROR SIGNALS

We propose a simple two-phase learning procedure, similar to the one proposed by Scellier & Bengio (2017). In the first phase of training, the inputs are set (clamped) to the input values. The state variable (all the other neurons) follows the dynamics Eq. 9 (or equivalently Eq. 13 with $\beta = 0$) and the output units are free. We call this phase the *free phase*, as the system relaxes freely towards the *free fixed point* $h^0$ without any external constraints on his output neurons. During this phase, the synaptic weights are unchanged.

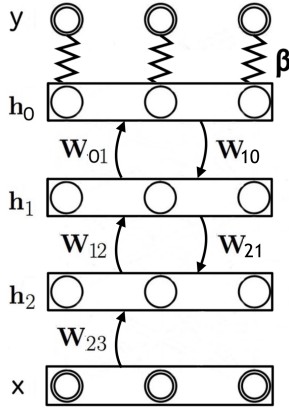
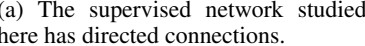
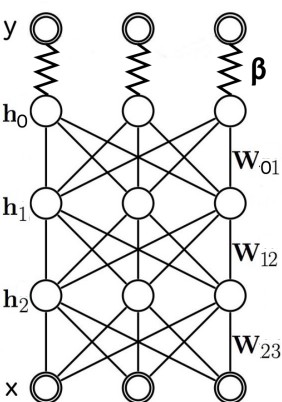

(a) The supervised network studied here has directed connections.

(b) In the framework of Equilibrium Propagation with the Hopfield energy, the network is assumed to have symmetric connections.

Figure 2: Input x is clamped. Neurons $h$ include "hidden layers" $h_2$ and $h_1$, and "output layer" $h_0$ that corresponds to the layer where the prediction is read. Target y has the same dimension as $h_0$. The clamping factor $\beta$ scales the "external force" $-\beta \frac{\partial C}{\partial h}$ that attracts the output $h_0$ towards the target y.

In the second phase, the influence parameter $\beta$ takes on a small positive value $\beta \gtrsim 0$. The state variable follows the dynamics Eq. 13 for that new value of $\beta$, and the synaptic weights follow the STDP-compatible weight change Eq. 5. This phase is referred to as the *weakly clamped phase*. The novel "external force" $-\beta \frac{\partial C}{\partial h}$ in the dynamics Eq. 13 acts on the output units and drives them towards their targets (Eq. 14). This force models the observation of y: it nudges the output units $h_0$ from their free fixed point value in the direction of their targets. Since this force only acts on the output layer $h_0$, the other hidden layers ($h_i$ with $i > 0$) are initially at equilibrium at the beginning of the weakly clamped phase. The perturbation caused at the output layer will then propagate backwards along the layers of the network, giving rise to "back-propagating" error signals. The network eventually settles to a new nearby fixed point, corresponding to the new value $\beta \gtrsim 0$, termed *weakly clamped fixed point* and denoted $h^\beta$.

### 4.4 VECTOR FIELD $\nu$ IN THE WEIGHT SPACE

Our model assumes that the STDP-compatible weight change (Eq. 5) occurs during the second phase of training (weakly clamped phase) when the network's state moves from the free fixed point $h^0$ to the weakly clamped fixed point $h^\beta$. Normalizing by a factor $\beta$ and letting $\beta \to 0$, we get the update rule $\Delta W \propto \nu(W)$ for the weights, where $\nu(W)$ is the vector defined as

$$\nu(W) := \frac{\partial \mu}{\partial W} \left( W, \text{x}, h^0 \right) \cdot \left. \frac{\partial h^\beta}{\partial \beta} \right|_{\beta=0}. \tag{16}$$

The vector $\nu(W)$ has the same dimension as $W$. Formally $\nu$ is a vector field in the weight space.

It is shown in section 5 that $\nu(W)$ is a proxy to the gradient $\frac{\partial J}{\partial W}$. The effectiveness of the proposed method is demonstrated through experimental studies (Figure 3).

## 5 THE VECTOR FIELD $\nu$ AS A PROXY FOR THE GRADIENT

In this section, we attempt to understand why the proposed algorithm is experimentally found to optimize the objective function $J$ (Figure 3). We say that $W$ is a "good parameter" if:

1. for any initial state for the neurons, the state dynamics $\frac{dh}{dt} = \mu(W, \text{x}, h)$ converges to a fixed point - a condition required for the algorithm to be correctly defined,

2. the scalar product $\frac{\partial J}{\partial W} \cdot \nu(W)$ at the point $W$ is negative - a desirable condition for the algorithm to optimize the objective function $J$.

Experiments show that the dynamics of $h$ (almost) always converges to a fixed point and that $J$ consistently decreases (Figure 3). This means that, during training, as the parameter $W$ follows the update rule $\Delta W \propto \nu(W)$, all values of $W$ that the network takes are "good parameters". In this section we attempt to explain why.

## 5.1 Explicit Formulas for $\frac{\partial J}{\partial W}$ and $\nu$

**Theorem 1.** *The gradient of $J$ can be expressed in terms of $\mu$ and $C$ as*

$$\frac{\partial J}{\partial W} = -\frac{\partial C}{\partial h} \cdot \left(\frac{\partial \mu}{\partial h}\right)^{-1} \cdot \frac{\partial \mu}{\partial W}. \tag{17}$$

*Similarly, the vector field $\nu$ (Eq. 16) is equal to*

$$\nu(W) = \frac{\partial C}{\partial h} \cdot \left(\left(\frac{\partial \mu}{\partial h}\right)^{T}\right)^{-1} \cdot \frac{\partial \mu}{\partial W}. \tag{18}$$

*In these expressions, all terms are evaluated at the fixed point $h^0$.*

Theorem 1 is proved in Appendix A. Note that the formulas show that $\nu(W)$ is related to $\frac{\partial J}{\partial W}$ and that the angle between these two vectors is directly linked to the "degree of symmetry" of the Jacobian of $\mu$.

An important particular case is the setting of Equilibrium Propagation (Scellier & Bengio, 2017), in which the vector field $\mu$ is a gradient field $\mu = -\frac{\partial E}{\partial h}$, meaning that it derives from an energy function $E$. In this case the Jacobian of $\mu$ is symmetric since it is the Hessian of $E$. Indeed $\frac{\partial \mu}{\partial h} = -\frac{\partial^2 E}{\partial h^2} = \left(\frac{\partial \mu}{\partial h}\right)^T$. Therefore, Theorem 1 shows that $\nu$ is also a gradient field, namely the gradient of the objective function $J$, that is $\nu = -\frac{\partial J}{\partial W}$. Note that in this setting the set of "good parameters" is the entire weight space - for all $W$, the dynamics $\frac{dh}{dt} = -\frac{\partial E}{\partial h}(W, h)$ converges to an energy minimum, and $W$ converges to a minimum of $J$ since $\Delta W \propto -\frac{\partial J}{\partial W}$.

We argue that the set of "good parameters" covers a large proportion of the weight space and that they contain the matrices $W$ that present a form of symmetry or "alignment". In the next subsection, we discuss how this form of symmetry may arise from the learning procedure itself.

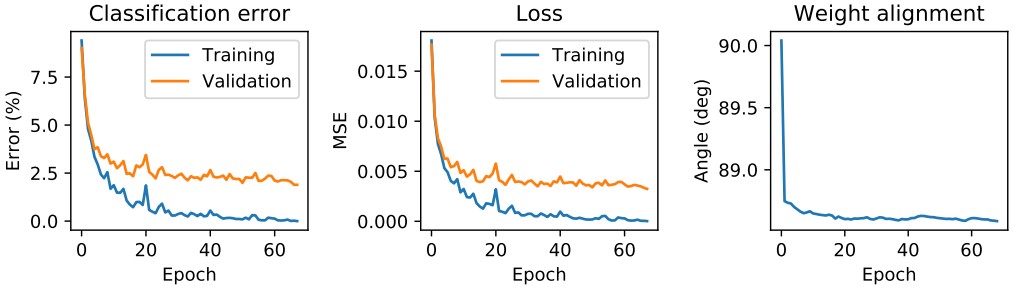

Figure 3: Example system trained on the MNIST dataset, as described in Appendix C. The objective function is optimized: the training error decreases to 0.00% in around 70 epochs. The generalization error is about 2%. Right: A form of symmetry or alignment arises between feedforward and feedback weights $W_{k,k+1}$ and $W_{k+1,k}$ in the sense that $tr(W_{k,k+1} \cdot W_{k+1,k}) > 0$. This architecture uses 3 hidden layers each of dimension 512.

## 5.2 A Form Of Symmetry Arises

Experiments show that a form of symmetry between feedforward and feedback weights arises from the learning procedure itself (Figure 3). Although the causes for this phenomenon aren't understood very well yet, it is worth pointing out that similar observations have been made in previous work and different settings.

A striking example is the following one. A major argument against the plausibility of backpropagation in feedforward nets is the weight transport problem: the signals sent forward in the network and those sent backward use the same connections. Lillicrap et al. (2016) have observed that, in the backward pass, (back)propagating the error signals through fixed random feedback weights (rather than the transpose of the feedforward weights) does not harm learning. Moreover, the learned feedforward weights $W_{k,k+1}$ tend to 'align' with the fixed random feedback weights $W_{k+1,k}$ in the sense that the trace of $W_{k,k+1} \cdot W_{k+1,k}$ is positive.

Denoising autoencoders without tied weights constitute another example of learning algorithms where a form of symmetry in the weights has been observed as learning goes on (Vincent et al., 2010).

The theoretical result from Arora et al. (2015) also shows that, in a deep generative model, the transpose of the generative weights perform approximate inference. They show that the symmetric solution minimizes the autoencoder reconstruction error between two successive layers of rectifying linear units.

## 6 POSSIBLE IMPLEMENTATION ON ANALOG HARDWARE

Our approach provides a basis for implementing machine learning models in continuous-time systems, while requirements regarding the actual dynamics are reduced to a minimum. This means that the model applies to a large class of physical realizations of vectorfield dynamics, including analog electronic circuits. Implementations of recurrent networks based on analog electronics have been proposed in the past, e.g. Hertz et al. (1997), however, these models typically required circuits and associated dynamics to adhere to an exact theoretical model. With our framework, we provide a way of implementing a learning system on a physical substrate without even knowing the exact dynamics or microscopic mechanisms that give rise to it. Thus, this approach can be used to analog electronic system end-to-end, without having to worry about exact device parameters and inaccuracies, which inevitably exist in any physical system. Instead of approximately implementing idealized computations, the actual analog circuit, with all its individual device variations, is trained to perform the task of interest. Thereby, the more direct implementation of the dynamics might result in advantages in terms of speed, power, and scalability, as compared to digital approaches.

## 7 CONCLUSION

Our model demonstrates that biologically plausible learning in neural networks can be achieved with relatively few assumptions. As a key contribution, in contrast to energy-based approaches such as the Hopfield model, we do not impose any symmetry constraints on the neural connections. Our algorithm assumes two phases, the difference between them being whether synaptic changes occur or not. Although this assumption begs for an explanation, neurophysiological findings suggest that phase-dependent mechanisms are involved in learning and memory consolidation in biological systems. Theta waves, for instance, generate neural oscillatory patterns that can modulate the learning rule or the computation carried out by the network Orr et al. (2001). Furthermore, synaptic plasticity, and neural dynamics in general, are known to be modulated by inhibitory neurons and dopamine release, depending on the presence or absence of a target signal. Frémaux & Gerstner (2016); Pawlak et al. (2010).

In its general formulation (Appendix A), the work presented in this paper is an extension of the framework of Scellier & Bengio (2017) to general dynamics. This is achieved by relaxing the requirement of an energy function. This generalization comes at the cost of not being able to compute the (true) gradient of the objective function but, rather a direction in the weight space which is related to it. Thereby, precision of the approximation of the gradient is directly related to the "alignment" between feedforward and feedback weights. Even though the exact underlying mechanism is not fully understood yet, we observe experimentally that during training the weights symmetrize to some extent, as has been observed previously in a variety of other settings (Lillicrap et al., 2016; Vincent et al., 2010; Arora et al., 2015). Our work shows that optimization of an objective function can be achieved without ever computing the (true) gradient. More thorough theoretical analysis needs to be carried out to understand and characterize the dynamics in the weight space that optimize objective functions. Naturally, the set of all such dynamics is much larger than the tiny subset of gradient-based dynamics.

Our framework provides a means of implementing learning in a variety of physical substrates, whose precise dynamics might not even be known exactly, but which simply have to be in the set of sup-

ported dynamics. In particular, this applies to analog electronic circuits, potentially leading to faster, more efficient, and more compact implementations.

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

# Appendix

## A   GENERAL FORMULATION

In this Appendix, we present the framework and the algorithm in their general formulations and we prove the theoretical results.

### A.1   PRELIMINARY DEFINITIONS

We consider a physical system specified by a state variable $s$ and a parameter variable $\theta$. The system is also influenced by an external input v, e.g. in the supervised setting $v = (x, y)$ where y is the target that the system wants to predict given x.

Let $s \mapsto \mu(\theta, v, s)$ be a vector field in the state space and $C(\theta, v, s)$ a cost function. We assume that the state dynamics induced by $\mu$ converges to a stable fixed point $s^0_{\theta, v}$, corresponding to the "prediction" from the model and characterized by

$$\mu\left(\theta, v, s^0_{\theta, v}\right) = 0. \tag{19}$$

The objective function that we want to optimize is the cost at the fixed point

$$J(\theta, v) := C\left(\theta, v, s^0_{\theta, v}\right). \tag{20}$$

Note the distinction between $J$ and $C$: the cost function is defined for any state $s$ whereas the objective function is the cost at the fixed point. The training objective (for a single data sample v) is

$$\text{find} \quad \arg\min_\theta J(\theta, v). \tag{21}$$

Equivalently, the training objective can be reformulated as a constrained optimization problem:

$$\text{find} \quad \arg\min_{\theta, s} C(\theta, v, s) \tag{22}$$

$$\text{subject to} \quad \mu(\theta, v, s) = 0, \tag{23}$$

where the constraint $\mu(\theta, v, s) = 0$ is the fixed point condition.

All traditional methods to compute the gradient of $J$ (adjoint method, implicit differentiation, Recurrent Backpropagation and Backpropagation Through Time or BPTT) are thought to be biologically implausible. Our approach is to give up on computing the gradient of $J$ and let the parameter variable $\theta$ follow a vector field $\nu$ in the parameter space which is "close" to the gradient of $J$.

Before defining $\nu$ we first introduce the "extended vector field"

$$\mu^\beta(\theta, v, s) := \mu(\theta, v, s) - \beta \, \frac{\partial C}{\partial s}(\theta, v, s), \tag{24}$$

where $\beta$ is a real-valued scalar called "influence parameter". Then we extend the notion of fixed point for any value of $\beta$. For any $\beta$ we define the $\beta$-fixed point $s^\beta_{\theta, v}$ such that

$$\mu^\beta\left(\theta, v, s^\beta_{\theta, v}\right) = 0. \tag{25}$$

Under mild regularity conditions on $\mu$ and $C$, the implicit function theorem ensures that, for a fixed data sample v, the funtion $(\theta, \beta) \mapsto s^\beta_{\theta, v}$ is differentiable.

Now for every $\theta$ and v we define the vector $\nu(\theta, v)$ in the parameter space as

$$\nu(\theta, v) := -\frac{\partial C}{\partial \theta}\left(\theta, v, s^0_{\theta, v}\right) + \frac{\partial \mu}{\partial \theta}\left(\theta, v, s^0_{\theta, v}\right) \cdot \left.\frac{\partial s^\beta_{\theta, v}}{\beta}\right|_{\beta=0}. \tag{26}$$

As shown in section 4, the second term on the right hand side can be estimated in a biologically realistic way thanks to a two-phase training procedure.

As compared to section 4, the definition of the vector $\nu(\theta, v)$ contains another term $-\frac{\partial C}{\partial \theta}\left(\theta, v, s_{\theta,v}^0\right)$ in the general case where the cost function $C$ also depends on the parameter $\theta$. This extra term can be measured in a biologically realistic way at the fixed point $s_{\theta,v}^0$ at the end of the free phase. For example if $C$ includes a regularization term such as $\frac{1}{2}\lambda \|\theta\|^2$, then $\nu(\theta, v)$ will include a backmoving force $-\lambda\theta$ modelling a form of synaptic depression.

## A.2 MAIN RESULT AND EXPLICIT FORMULAS

**Lemma 2.** *Let $s \mapsto \mu^\beta(\theta, s)$ be a differentiable vector field, and $s_\theta^\beta$ a fixed point characterized by*

$$\mu^\beta\left(\theta, s_\theta^\beta\right) = 0. \tag{27}$$

*Then the partial derivatives of the fixed point are given by*

$$\frac{\partial s_\theta^\beta}{\partial \theta} = -\left(\frac{\partial \mu^\beta}{\partial s}\left(\theta, s_\theta^\beta\right)\right)^{-1} \cdot \frac{\partial \mu^\beta}{\partial \theta}\left(\theta, s_\theta^\beta\right) \tag{28}$$

*and*

$$\frac{\partial s_\theta^\beta}{\partial \beta} = -\left(\frac{\partial \mu^\beta}{\partial s}\left(\theta, s_\theta^\beta\right)\right)^{-1} \cdot \frac{\partial \mu^\beta}{\partial \beta}\left(\theta, s_\theta^\beta\right). \tag{29}$$

*Proof of Lemma 2.* First we differentiate the fixed point equation Eq. 27 with respect to $\theta$:

$$\frac{d}{d\theta}(27) \Rightarrow \frac{\partial \mu^\beta}{\partial \theta}\left(\theta, s_\theta^\beta\right) + \frac{\partial \mu^\beta}{\partial s}\left(\theta, s_\theta^\beta\right) \cdot \frac{\partial s_\theta^\beta}{\partial \theta} = 0. \tag{30}$$

Rearranging the terms we get Eq. 28. Similarly we differentiate the fixed point equation Eq. 27 with respect to $\beta$:

$$\frac{d}{d\beta}(27) \Rightarrow \frac{\partial \mu^\beta}{\partial \beta}\left(\theta, s_\theta^\beta\right) + \frac{\partial \mu^\beta}{\partial s}\left(\theta, s_\theta^\beta\right) \cdot \frac{\partial s_\theta^\beta}{\partial \beta} = 0. \tag{31}$$

Rearranging the terms we get Eq. 29. $\qquad \square$

**Theorem 3.** *The gradient of the objective function is equal to*

$$\frac{\partial J}{\partial \theta} = \frac{\partial C}{\partial \theta} - \frac{\partial C}{\partial s} \cdot \left(\frac{\partial \mu}{\partial s}\right)^{-1} \cdot \frac{\partial \mu}{\partial \theta} \tag{32}$$

*and the vector field $\nu$ is equal to*

$$\nu = -\frac{\partial C}{\partial \theta} + \frac{\partial C}{\partial s} \cdot \left(\left(\frac{\partial \mu}{\partial s}\right)^T\right)^{-1} \cdot \frac{\partial \mu}{\partial \theta}. \tag{33}$$

*All the factors on the right-hand sides of Eq. 32 and Eq. 33 are evaluated at the fixed point $s_\theta^0$.*

*Proof of Theorem 3.* Let us compute the gradient of the objective function with respect to $\theta$. Using the chain rule of differentiation we get

$$\frac{\partial J}{\partial \theta} = \frac{\partial C}{\partial \theta} + \frac{\partial C}{\partial s} \cdot \frac{\partial s_\theta^0}{\partial \theta}. \tag{34}$$

Hence Eq. 32 follows from Eq. 28 evaluated at $\beta = 0$. Similarly, the expression for the vector field $\nu$ (Eq. 33) follows from its definition (Eq. 26), the identity Eq. 29 evaluated at $\beta = 0$ and the fact that $\frac{\partial \mu^\beta}{\partial \beta} = -\frac{\partial C}{\partial s}$. $\qquad \square$

We finish by stating and proving a last result. Consider the setting introduced in section 4 with the quadratic cost function $C = \frac{1}{2}\|y - h_0\|^2$. In the weakly clamped phase, the "external influence" $-\beta(y - h_0)$ added to the vector field $\mu$ (with $\beta \gtrsim 0$) slightly attracts the output state $h_0$ to the target $y$. It is intuitively clear that the weakly clamped fixed point is better than the free fixed point in terms of prediction error. Proposition 5 below generalizes this property to any vector field $\mu$ and any cost function $C$.

**Proposition 4.** *Let $s^0$ be a stable fixed point of the vector field $s \mapsto \mu(s)$, in the sense that $(s - s^0) \cdot \mu(s) < 0$ for $s$ in the neighborhood of $s^0$ (i.e. the vector field at $s$ points towards $s^0$). Then the Jacobian of $\mu$ at the fixed point $\frac{\partial \mu}{\partial s}(s^0)$ is negative, in the sense that*

$$\forall v, \qquad v \cdot \frac{\partial \mu}{\partial s}(s^0) \cdot v \leq 0. \tag{35}$$

*Proof.* Let $v$ be a vector in the state space, $\alpha > 0$ a positive scalar and let $s := s^0 + \alpha v$. For $\alpha$ small enough, the vector $s$ is in the region of stability of $s^0$. Using a first order Taylor expansion and the fixed point condition $\mu(s^0) = 0$ we get

$$0 > (s - s^0) \cdot \mu(s) \tag{36}$$

$$= \alpha v \cdot \mu(s^0 + \alpha v) \tag{37}$$

$$= \alpha v \cdot \frac{\partial \mu}{\partial s}(s^0) \cdot \alpha v + o(\alpha^2) \tag{38}$$

as $\alpha \to 0$. Hence the result. $\qquad\square$

The following proposition shows that, unless the free fixed point $s^0_{\theta,\mathrm{v}}$ is already optimal in terms of cost value, for $\beta > 0$ small enough, the nudged fixed point $s^\beta_{\theta,\mathrm{v}}$ achieves lower cost value than the free fixed point. Thus, a small perturbation due to a small increment of $\beta$ nudges the network towards a configuration that reduces the cost value.

**Proposition 5.** *Let $s^\beta_\theta$ be a stable fixed point of the extended vector field $\mu^\beta = \mu - \beta \frac{\partial C}{\partial s}$. Then the derivative of the function*

$$\beta \mapsto C\left(\theta, s^\beta_\theta\right) \tag{39}$$

*at $\beta = 0$ is non-positive.*

*Proof of Proposition 5.* Multiplying both sides of Eq. 31 on the left by $-\left(\frac{\partial s^\beta_\theta}{\partial \beta}\right)^T$ and rearranging the terms, we get

$$-\left(\frac{\partial s^\beta_\theta}{\partial \beta}\right)^T \cdot \frac{\partial \mu^\beta}{\partial \beta} = \left(\frac{\partial s^\beta_\theta}{\partial \beta}\right)^T \cdot \frac{\partial \mu^\beta}{\partial s} \cdot \frac{\partial s^\beta_\theta}{\partial \beta} \leq 0. \tag{40}$$

The inequality holds because $\frac{\partial \mu^\beta}{\partial s}\left(\theta, s^\beta_\theta\right)$ is negative as $s^\beta_\theta$ is a stable fixed point of $\mu^\beta$ (Eq. 35). Since $\frac{\partial \mu^\beta}{\partial \beta} = -\frac{\partial C}{\partial s}$, the left-hand side, for $\beta = 0$, represents the derivative of

$$\beta \mapsto C\left(\theta, s^\beta_\theta\right). \tag{41}$$

$\qquad\square$

## B ADJOINT METHOD AND RELATED ALGORITHMS

Earlier work have proposed various methods to compute the gradient of the objective function $J$ (Eq. 20). One common method is the "adjoint method". In the context of fixed point recurrent neural networks studied here, the adjoint method leads to Backpropagation Through Time (BPTT) and "Recurrent Backpropagation" (Almeida, 1987; Pineda, 1987). BPTT is the method of choice today for deep learning but its biological implausibility is obvious - it requires the network to store all its past states for the propagation of error derivatives in the second phase. Recurrent Backpropagation corresponds to a special case of BPTT where the network is initialized at the fixed point. This algorithm does not need to store the past states of the network (the state at the fixed point suffices) but it still requires neurons to send a different kind of signals through a different computational path in the second phase, which seems therefore less biologically plausible than our algorithm.

Our approach is to give up on the idea of computing the *gradient* of the objective function. Instead we show that the STDP-compatible weight change computes a proxy to the gradient in a more biologically plausible way.

## B.1 CONTINUOUS-TIME BACKPROPAGATION

For completeness, we state and prove a continuous-time version of Backpropagation Through Time and Recurrent Backpropagation. The formulas for the propagation of error derivatives (Theorem 6 and Corollary 7) will make it obvious that our algorithm is more biologically plausible than both of these algorithms.

To keep notations simple, we omit to write dependences in the data sample v. We consider the dynamics $\frac{ds}{dt} = \mu(\theta, s)$ and denote by $s_t$ the state of the system at time $t \geq 0$ when it starts from an initial state $s_0$ at time $t = 0$. Note that $s_t$ converges to the fixed point $s_\theta^0$ as $t \to \infty$. We then define a family of objective functions

$$L(\theta, s_0, T) := C(\theta, s_T) \tag{42}$$

for every couple $(s_0, T)$ of initial state $s_0$ and duration $T \geq 0$. This is the cost of the state at time $t = T$ when the network starts from the state $s_0$ at time $t = 0$. Note that $L(\theta, s_0, T)$ tends to $J(\theta)$ as $T \to \infty$ (Eq. 20).

We want to compute the gradient $\frac{\partial L}{\partial \theta}(\theta, s_0, T)$ as $T \to \infty$. For that purpose, we fix $T$ to a large value and we consider the following quantity

$$\frac{\partial L}{\partial s_{T-t}} := \frac{\partial L}{\partial s}(\theta, s_{T-t}, t), \tag{43}$$

which represents the "partial derivative of the cost with respect to the state at time $T - t$". In other words this is the "partial derivative of the cost with respect to the $(T - t)$-th hidden layer" if one regards the network as unfolded in time (though time is continuous here). The formulas in Theorem 6 below correspond to a continuous-time version of BPTT for the propagation of the partial derivatives $\frac{\partial L}{\partial s_{T-t}}$ backward in time.

**Theorem 6** (Continuous-Time Backpropagation Through Time). *The process of "partial derivatives"* $\frac{\partial L}{\partial s_{T-t}}$ *is such that*

$$\frac{d}{dt}\frac{\partial L}{\partial s_{T-t}} = \left(\frac{\partial \mu}{\partial s}(\theta, s_{T-t})\right)^T \cdot \frac{\partial L}{\partial s_{T-t}}, \tag{44}$$

*and the gradient* $\frac{\partial L}{\partial \theta}(\theta, s_{T-t}, t)$ *is such that*

$$\frac{d}{dt}\frac{\partial L}{\partial \theta}(\theta, s_{T-t}, t) = \left(\frac{\partial \mu}{\partial \theta}(\theta, s_{T-t})\right)^T \cdot \frac{\partial L}{\partial s_{T-t}}. \tag{45}$$

Computing $\frac{\partial L}{\partial s_{T-t}}$ and $\frac{\partial L}{\partial \theta}(\theta, s_{T-t}, t)$ thanks to Eq. 44 and Eq. 45 is biologically infeasible since it requires storing the past states $s_{T-t}$.

In the particular case where the network is initialized at the fixed point, then we have $s_{T-t} = s_\theta^0$ for all $t$ and we get a continuous-time version of "Recurrent Backpropagation" (Almeida, 1987; Pineda, 1987).

**Corollary 7** (Continuous-Time Recurrent Backpropagation). *The process* $\frac{\partial L}{\partial s}(\theta, s_\theta^0, t)$ *for* $t \geq 0$ *satisfies the differential equation*

$$\frac{d}{dt}\frac{\partial L}{\partial s}(\theta, s_\theta^0, t) = \left(\frac{\partial \mu}{\partial s}(\theta, s_\theta^0)\right)^T \cdot \frac{\partial L}{\partial s}(\theta, s_\theta^0, t). \tag{46}$$

*and the process* $\frac{\partial L}{\partial \theta}(\theta, s_\theta^0, t)$ *for* $t \geq 0$ *satisfies*

$$\frac{d}{dt}\frac{\partial L}{\partial \theta}(\theta, s_\theta^0, t) = \left(\frac{\partial \mu}{\partial \theta}(\theta, s_\theta^0)\right)^T \cdot \frac{\partial L}{\partial s}(\theta, s_\theta^0, t). \tag{47}$$

*Here the notation* $\frac{\partial L}{\partial \theta}$ *represents the partial derivative with respect to the first argument, which does not include the path through* $s_\theta^0$.

Recurrent Backpropagation does not require the state $s$ go backward in time in the second phase. The state of the network stays at the fixed point $s_\theta^0$. However we still need a special computational path for the computation of $\frac{\partial L}{\partial s}(\theta, s_\theta^0, t)$. From the point of view of biological plausibility, it is not clear how this can be done and how the transpose of the Jacobian $\left(\frac{\partial \mu}{\partial s}(\theta, s_\theta^0)\right)^T$ can be measured.

*Proof of Theorem 6.* To keep notations simple, we omit to write the dependence in $\theta$. First we show that for all $s$ and $t$ we have

$$\frac{\partial L}{\partial t}(s,t) = \frac{\partial L}{\partial s}(s,t) \cdot \mu(s). \tag{48}$$

To this end note that

$$L(s_u, t-u) = L(s_0, t) \tag{49}$$

is independent of $u$. Therefore

$$\frac{d}{du} L(s_u, t-u) = 0 \tag{50}$$

$$= -\frac{\partial L}{\partial t}(s_u, t-u) + \frac{\partial L}{\partial s}(s_u, t-u) \cdot \mu(s_u). \tag{51}$$

Here we have used the chain rule of differentiation and the differential equation of motion. Evaluating this expression for $u = 0$ we get Eq. 48 since the initial point $s_0$ is arbitrary. Then, differentiating Eq. 48 with respect to $s$, we get

$$\frac{\partial^2 L}{\partial t \partial s}(s,t) = \frac{\partial^2 L}{\partial s^2}(s,t) \cdot \mu(s) + \left(\frac{\partial \mu}{\partial s}(s)\right)^T \cdot \frac{\partial L}{\partial s}(s,t) = 0. \tag{52}$$

Now let us differentiate $\frac{\partial L}{\partial s}(s_{T-t}, t)$ with respect to $t$. Using the chain rule of differentiation, the differential equation of motion and Eq. 52 (at the point $s = s_{T-t}$) we get

$$\frac{d}{dt} \frac{\partial L}{\partial s}(s_{T-t}, t) \tag{53}$$

$$= \frac{\partial^2 L}{\partial t \partial s}(s_{T-t}, t) - \frac{\partial^2 L}{\partial s^2}(s_{T-t}, t) \cdot \mu(s_{T-t}) \tag{54}$$

$$= \left(\frac{\partial \mu}{\partial s}(s_{T-t})\right)^T \cdot \frac{\partial L}{\partial s}(s_{T-t}, t). \tag{55}$$

Hence Eq. 44. We derive Eq. 45 similarly by differentiating $\frac{\partial L}{\partial \theta}(\theta, s_{T-t}, t)$ with respect to $t$. $\qquad \square$

## C  IMPLEMENTATION DETAILS OF THE MODEL

Our model is a recurrently connected neural network without any constraint on the feedback weight values. We train multi-layered networks with 2 or 3 hidden layers, with no skip-layer connections and no lateral connections within layers.

Rather than doing the weight updates at all time steps, we use a single update at the end of the weakly clamped phase:

$$\Delta W \propto \frac{\partial \mu}{\partial h}\left(h^0\right) \cdot \frac{h^\beta - h^0}{\beta}. \tag{56}$$

The prediction is made on the last layer at the free fixed point $h_0^0$ at the end of the first phase relaxation. The predicted value $h_{\text{pred}}$ is the index of the output unit whose activation is maximal among the 10 output units:

$$h_{\text{pred}} := \arg\max_i h_{0,i}^0. \tag{57}$$

**Implementation of the differential equation of motion.** We start by clamping x to the data values. Then, to implement Eq. 13, we use the Euler method. We discretize time into short time lapses of duration $\epsilon$ and update the state variable $h$ thanks to the following equation:

$$h \leftarrow h - \epsilon \mu^\beta(W, \text{x}, h, \text{y}). \tag{58}$$

For our experiments, we choose the hard sigmoid activation function $\rho(h_i) = 0 \vee h_i \wedge 1$, where $\vee$ denotes the max and $\wedge$ the min. For this choice of $\rho$, since $\rho'(h_i) = 0$ for $h_i < 0$, it follows from Eq. 1 and Eq. 14 that if $h_i < 0$ then $\frac{\partial F}{\partial h_i}(\theta, \text{v}, \beta, s) = -h_i > 0$. This force prevents the hidden unit $h_i$ from going in the range of negative values. The same is true for the output units. Similarly,

$h_i$ cannot reach values above 1. As a consequence $h_i$ must remain in the domain $0 \leq h_i \leq 1$. Therefore, rather than the standard gradient descent (Eq. 58), we will use a slightly different update rule for the state variable $h$:

$$h \leftarrow 0 \vee h - \epsilon \mu^\beta (W, \mathrm{x}, h, \mathrm{y}) \wedge 1. \tag{59}$$

This little implementation detail turns out to be very important: if the $i$-th hidden unit was in some state $h_i < 0$, then Eq. 58 would give the update rule $h_i \leftarrow (1 - \epsilon)h_i$, which would imply again $h_i < 0$ at the next time step (assuming $\epsilon < 1$). As a consequence $h_i$ would remain in the negative range forever.

We use different learning rates for the different layers in our experiments. We do not have a clear explanation for why this improves performance, but we believe that this is due to the finite precision with which we approach the fixed points.

The hyperparameters chosen for each model are shown in Table 1 and the results are shown in Figure 3. We initialize the weights according to the Glorot-Bengio initialization (Glorot & Bengio, 2010). For efficiency of the experiments, we use minibatches of 20 training examples.

| Architecture | Iterations (first phase) | Iterations (second phase) | $\epsilon$ | $\beta$ | $\alpha_1$ | $\alpha_2$ | $\alpha_3$ | $\alpha_4$ |
|---|---|---|---|---|---|---|---|---|
| $784 - 512 - 512 - 10$ | 200 | 100 | 0.001 | 1.0 | 0.4 | 0.1 | 0.01 | $--$ |
| $784 - 512 - 512 - 512 - 10$ | 200 | 100 | 0.001 | 1.0 | 1.0 | 0.1 | 0.04 | 0.002 |

Table 1: Hyperparameters. for both the 2 and 3 layered MNIST. Example system trained on the MNIST dataset, as described in Appendix C. The objective function is optimized: the training error decreases to 0.00%. The generalization error lies between 2% and 3% depending on the architecture. The learning rate $\epsilon$ is used for iterative inference (Eq. 59). $\beta$ is the value of the clamping factor in the second phase. $\alpha_k$ is the learning rate for updating the parameters in layer $k$.

We were also able to train on MNIST using a Convolutional Neural Network (CNN). We got around 2% generalization error. The hyperparameters chosen to train this Convolutional Neural Network are shown in Table 2.

| Operation | Kernel | Strides | Feature Maps | Non Linearity |
|---|---|---|---|---|
| Convolution | 5 x 5 | 1 | 32 | Relu |
| Convolution | 5 x 5 | 1 | 64 | Relu |

Table 2: Hyperparameters for MNIST CNN experiments.

