# OpenReview forum: "Extending the Framework of Equilibrium Propagation to General Dynamics"
_ICLR.cc/2018/Conference — Invite to Workshop Track_

### Official Review · AnonReviewer3 · 2017-11-27
**An interesting extension of the equilibrium propagation algorithm, but the results seem to be incomplete and necessary additional analyses are missing.**

**Rating:** 4
**Confidence:** 4

**Review:**

The manuscript discusses a learning algorithm that is based on the equilibrium propagation method, which can be applied to networks with asymmetric connections. This extension is interesting, but the results seem to be incomplete and missing necessary additional analyses. Therefore, I do not recommend acceptance of the manuscript in its current form. The main issues are:

1) The theoretical result is incomplete since it fails to show that the algorithm converges to a meaningful learning result. Also the experimental results do not sufficiently justify the claims.

2) The paper further makes statements about the performance and biological plausibility of the proposed algorithm that do not hold without additional justification.

3) The paper does not sufficiently discuss and compare the relevant neuroscience literature and related work.

Details to major points:

1) The presentation of the theoretical results is misleading. Theorem 1 shows that the proposed neuron dynamics has a fixed point that coincides with a local minimum of the objective function if the weights are symmetric. However, this was already clear from the original equilibrium propagation paper. The interesting question is whether the proposed algorithm automatically converges to the condition of symmetric weights, which is left unanswered. In Figure 3 experimental evidence is provided, but the results are not convincing given that the weight alignment only improves by ~1° throughout learning (compared to >45° in Lillicrap et al., 2017). It is even unclear to me if this effect is statistically significant. How many trials did the authors average over here? The authors should provide standard statistical significance measures for this plot. Since no complete theoretical guarantees are provided, a much broader experimental study would be necessary to justify the claims made in the paper.

2) Throughout the paper it is claimed that the proposed learning algorithm is biologically plausible. However, this argument is also not sufficiently justified. Most importantly, it is unclear how the proposed algorithm would behave in a biologically realistic recurrent networks and it is unclear how the different learning phases should be realized in the brain.

Neural networks in the brain are abundantly recurrent. Even in the layered structure of the neocortex one finds dense lateral connectivity between neurons on each layer. It is not clear to me how the proposed algorithm could be applied to such networks. In a recurrent network, rolled-out over time, information would need to be passed forward and backwards in time. The proposed algorithm does not seem to provide a solution to this temporal credit assignment problem. Also in the experiments the algorithm is applied only to feedforward architectures. What would happen if recurrent networks were used to learn temporal tasks like TIMIT? Please discuss.

In the discussion on page 8 the authors further argue that the learning phases of the proposed algorithm could be implemented in the cortex through theta waves that modulate long-term plasticity. To support this theory the authors cite the results from Orr et al., 2001, where hippocampal place cells in behaving rats were studied. To my knowledge there is no consensus on the precise nature of this modulation of plasticity. E.g. in Wyble et al. 2003, it was observed that application of learning protocols at different phases of theta waves actually leads to a sign change in learning, i.e. long term potentiation was modulated to depression. It seems to me that the algorithm is not compatible with these other experimental findings, since gradients only point in the correct direction towards the final phase and any non-zero learning rate in other phases would therefore perturb learning. Did the authors try non-optimal learning rate schedules in the experiments (including sign change etc.) to test the robustness of the proposed algorithm? Also to my knowledge, the modulatory effect of theta rhythms has so far only been described in the CA1 region of rodent hippocampus which is a very specialized region of the brain (see Hanslmayr et al., 2016, for a review and a modern hypothesis on the role of theta rhythms in the brain).

Furthermore, the discussion of the possible implementation of the learning algorithm in analog hardware on page 8 is missing an explanation of how the different learning phases of the algorithm are controlled on the chip. One of the advantages of analog hardware is that it does not require global clocking, unlike classical digital hardware, which is expensive in wiring and energy requirement. It seems to me that this advantage would disappear if the algorithm was brought to an analog chip, since global information about the learning phase has to be communicated to each synapse. Is there an alternative to a global wiring scheme to convey this information throughout the whole chip? Please discuss this in more depth.

3) The authors apply the learning algorithm only to the MNIST dataset, which is a relatively simple task. Similar results were also achieved using random feedback alignment (Lillicrap et al., 2017). Also, the evolutionary strategies method (Salimans et al., 2017), was recently used for learning deep networks and applied to complex reinforcement learning problems and could likewise also be applied to simple classification tasks. Both these methods are arguably as simple and biologically plausible as the proposed algorithm. It would be good to try other standard benchmark tasks and report and compare the performance there. Furthermore, the paper is missing a broader related work section that discusses approaches for biologically plausible learning rules for deep neural architectures.


Minor points:

The proposed algorithm uses different learning rates that shrink exponentially with the layer number. Have the authors explored whether the algorithm works for really deep architectures with several tens of layers? It seems to me that the used learning rate heuristic may hinder scalability of equilibrium propagation.

On page 5 the authors write: "However we observe experimentally that the dynamics almost always converges." This needs to be quantified. Did the authors find that the algorithm is very sensitive to initial conditions?


References:

Bradley P. Wyble, Vikas Goyal, Christina A. Rossi, and Michael E. Hasselmo. Stimulation in Hippocampal Region CA1 in Behaving Rats Yields Long-Term Potentiation when Delivered to the Peak of Theta and Long-Term Depression when Delivered to the Trough James M. Hyman. Journal of Neuroscience. 2003.

Simon Hanslmayr, Bernhard P. Staresina, and Howard Bowman. Oscillations and Episodic Memory: Addressing the Synchronization/Desynchronization Conundrum. Trends in Neurosciences. 2016.

Tim Salimans, Jonathan Ho, Xi Chen, Szymon Sidor, Ilya Sutskever. Evolution Strategies as a Scalable Alternative to Reinforcement Learning. Arxiv. 2017.

---

> ### Author Response · Authors · 2018-01-05
> **Response to Reviewer 3**
>
> We thank the reviewer for their thorough review.
>
> We agree with each of the points 1, 2, and 3.
>
> 1) The plot corresponds to a single trial (but different trials typically always show the same curve). We agree that the alignment effect is not very convincing. However, further experiments which we have carried out in the meantime show that the objective function J always decreases, even when the weights are totally misaligned (e.g. when one initializes the feedback weights W_ji to be the opposite of the feedforward weights W_ij, that is W_ji = -W_ij)
>
> 2) Regarding the objection concerning the abundance of lateral connections in biological networks, note that our theory also applies to neural architectures that include lateral connections (although for simplicity of presentation we have considered the case of multi-layer networks with neither lateral nor skip-layer connections).
>
> Our algorithm only applies to the standard supervised scenario where one predicts y given x. It is unclear how to extend the theory to sequential data.
>
> Regarding the possible implementation on analog circuits, the way we conceive it is that global clocking for switching phases would be done digitally. Only the phases themselves would be performed analogically.
>
> Our algorithm does not scale well with the number of layers (yet!), both because of the learning rates and because of the lengthy "free relaxation phase".

---

### Official Review · AnonReviewer2 · 2017-11-27

**Rating:** 3
**Confidence:** 4

**Review:**

tl;dr: The paper extends equilibrium propagation to recurrent networks, but doesn't test the algorithm on a dataset requiring a recurrent architecture.

The experimental results are extremely weak, just for MNIST. There are two problems with this. Firstly, the usual issues with MNIST being too easy, idiosyncratic in many ways, and over-studied. It is a good sanity check but not enough for an ICLR paper. Secondly, and more importantly, MNIST does not require a recurrent architecture. Applying an RNN to MNIST (as opposed to, say, permuted MNIST) is a strange thing to do. The authors should investigate datasets with sequential structure. There *tons* of examples in audio, language, etc.

As a consequence of the extremely limited experiments, it is difficult to know how much to trust the papers claims (top of page 5, top of page 7, near the end of page 8) about the algorithm optimizing the objective “experimentally”. Yes, it does so for MNIST. What about in more difficult cases?

Detailed comments:
“We use different learning rates for the different layers in our experiments. We do not have a clear explanation for why this improves performance ...” Introducing an additional hyperparameter per layer is a major drawback of the approach.

---

> ### Author Response · Authors · 2018-01-05
> **Response to Reviewer 2**
>
> Thanks for reviewing our submission.
>
> Our algorithm does not apply to sequential data. As explained throughout the paper, we are interested in the standard supervised setting (predicting y given x).
>
> Our algorithm is not an extension of equilibrium propagation to RNN - the original algorithm of equilibrium propagation is already a recurrent model (with fixed input). Recurrent model does not necessarily mean sequential input data - biological networks are recurrent networks and they work in a recurrent manner even when presented with fixed input signals.
>
> Our contribution is not to beat a benchmark on standard machine learning datasets, but to propose a learning algorithm similar in spirit to backpropagation and more faithful to current knowledge in neuroscience.

---

### Official Review · AnonReviewer4 · 2017-12-15

**Rating:** 6
**Confidence:** 2

**Review:**

The paper proposes a new learning algorithm for learning neural networks that may be biologically plausible. The paper builds upon the paper by Scellier & Bengio but doesn't assume symmetric weights. I couldn't judge how solid the biological plausibility argument but my understanding is that there is no universal agreement in neuroscience about it so I would tend to be open to most of the suggestions. As a non-expert in this field, I found this result of this paper pretty interesting, given experimentally the algorithm does work well for MNIST (which is already interesting to me, given the limited progress in this area).

---

> ### Author Response · Authors · 2018-01-05
> **Response to Reviewer 4**
>
> We thank the reviewer for their remarks.
>
> It is true that there is little progress in biologically plausible "backpropagation" in general. Our work is a small step towards achieving this goal.

---

### Decision · Program_Chairs · 2018-01-29
**ICLR 2018 Conference Acceptance Decision**

**Decision:**

Invite to Workshop Track

**Comment:**

 + interesting novel extension of equilibrium propagation, as a biologically more plausible alternative to  backpropagation, with encouraging initial experimental validation.
 - currently lacks theoretical guarantees regarding convergence of the algorithm to a meaningful result
 - experimental study should be more extensive to support the claims